# Insecticidal Effect of the Entomopathogenic Fungus *Lecanicillium araneicola* HK-1 in *Aphis craccivora* (Hemiptera: Aphididae)

**DOI:** 10.3390/insects14110860

**Published:** 2023-11-06

**Authors:** Shengke Liu, Jinhua Li, Qing Feng, Linglong Chu, Zhiqiong Tan, Xuncong Ji, Pengfei Jin

**Affiliations:** 1Key Laboratory of Green Prevention and Control of Tropical Plant Diseases and Pests, Ministry of Education and School of Plant Protection, Hainan University, Haikou 570228, China; l13226379151@163.com (S.L.); lijinhua0801@126.com (J.L.); chulinglonghn@163.com (L.C.); 2Key Laboratory of Plant Disease and Pest Control of Hainan Province, Haikou 570228, China; fengqing@hnaas.org.cn; 3School of Life Sciences, Hainan University, Haikou 570228, China; tanzhiqiong2002@163.com

**Keywords:** *Lecanicillium*, *Aphis craccivora*, infection process, crude extract, enzyme system

## Abstract

**Simple Summary:**

*Aphis craccivora* is an important pest affecting various crops worldwide. *Lecanicillium* spp. are important entomopathogenic fungi of Hypocreales. However, there have been few studies on the infection of *A. craccivora* by *Lecanicillium* and the related insecticidal mechanisms. In this study, *Lecanicillium araneicola* HK-1 was shown to produce protease and chitinase to degrade the integument of *A. craccivora*, resulting in insect’s death because of the synergistic effects of parasitism and harmful metabolites of the strain HK-1.

**Abstract:**

*Aphis craccivora* (Hemiptera: Aphididae) is an important pest affecting various crops worldwide. However, only few studies have been conducted on the infection of *A. craccivora* by *Lecanicillium* and related insecticidal mechanisms. We investigated the infection process of *A. craccivora* by *Lecanicillium araneicola* HK-1 using fluorescence microscopy and scanning electron microscopy (SEM), and our results indicated that the conidia of strain HK-1 easily attached to the feet and dorsum of *A. craccivora*. The activities of chitinase and extracellular protease were induced in the aphid after treatment with HK-1. A bioassay on *A. craccivora* showed that the median lethal concentration (LC_50_) of the fungus crude extract was 24.00 mg mL^−1^ for 24 h of treatment. Additionally, the results showed that the crude extract disrupted the enzyme system of *A. craccivora*, inducing the inhibition of carboxylesterase (CarE) and the induction of glutathione S-transferase (GST) and acetylcholinesterase (AChE). Combining these results with those of a gas chromatography–mass spectrometry (GC-MS) analysis, it is suggested that *p*-cymene, hymecromone, 9,12-octadecadienoic acid (Z, Z) methyl ester, and 9,12-octadecadienoic acid (Z, Z) may be connected to the insecticidal effects we observed. This study provides a theoretical basis for the use of *L. araneicola* HK-1 as a potential biological control agent.

## 1. Introduction

*Aphis craccivora* (Hemiptera: Aphididae) is one of the most destructive pests worldwide [1]. It can infest a variety of plants, including cowpeas, alfalfa, and peanuts, and sometimes causes 20–40% yield losses [2,3]. Adult aphids and nymphs prefer sucking tender leaves, shoots, and stems, which leads to leaf curling, leaf yellowing, stem wilting, and even plant death [4]. Additionally, the honeydew produced by aphids can hinder photosynthesis in plants and cause the development of sooty molds [5]. Furthermore, aphids are involved in the spread of various destructive plant viruses [6], including *barley yellow dwarf virus*, *papaya ring spot virus*, and *watermelon mosaic virus*, which reduce agricultural production and cause large economic losses worldwide each year [7,8,9].

Insecticide application remains the primary method of aphid control in field crops and is based on the use of pyrethroids, organophosphorus compounds, carbamates, etc. [10]. However, the frequent use and abuse of chemical pesticides results in insect resistance, environmental pollution, soil deterioration, and high levels of pesticide residues [11,12]. Therefore, it is imperative to develop safe, effective, and sustainable biological methods for aphid control. The application of entomopathogenic fungi is a promising alternative approach for aphid control because of the low cost, environmental friendliness, innocuous nature to non-target organisms of these fungi and the wide range of raw material sources available [13].

Generally, the conidia of entomopathogenic fungi can colonize the pest body surface and develop into germ tubes and appressoria at suitable temperatures and humidity levels [14,15]. Entomopathogenic fungi can produce substantial amounts of proteases and chitinases that degrade the host body wall, as the insect cuticle is mainly composed of chitin and proteins [16]. Subsequently, the fungal hyphae penetrate the cuticle using mechanical pressure, and the invaginating hyphae erode and degrade the neighboring cuticles [17,18]. The insect immune system is destroyed by the fungal hyphae, which multiply and destroy the tissues and organs of the host, causing its death [19].

The fungus *Lecanicillium*, originally isolated from *Lecanii coffeae* in Ceylon by Nivter in 1861, is an important entomopathogenic fungus of the Hypocreales that is used as a biocontrol agent for pests, including thrips, aphids, and whiteflies [20]. The infection process of *Lecanicillium* in some aphids was reported; however, it differs with the type of aphid. Ganassi et al. reported that *L. lecanii* adhered randomly to all regions of the *Schizaphis graminum* body surface without any apparent preference [21]. In contrast, Zhang et al. showed that *L. longisporum* colonized weak areas of *Myzus persicae*, including the compound eye, antennal socket, base of the legs, and genital segments [22]. Secondary metabolites play a major role in insect infection by entomopathogenic fungi [23]. Entomopathogenic fungi release immunosuppressive secondary metabolites, including destruxin, beauverolides, and cytochalasins [24]. However, most publications focused on crude extracts, and few reported the detailed entomopathogenic mechanisms of *Lecanicillium* in aphids based on the study of the infection process and the produced metabolites. Therefore, further research on the isolation and purification of insecticidally active substances from *Lecanicillium* is required.

Insect enzymes such as carboxylesterase (CarE) and glutathione S-transferase (GST) metabolize harmful exogenous substances [25], and the enzyme acetylcholinesterase (AChE) can degrade acetylcholine, terminating the excitatory effect of neurotransmitters on the post-synaptic membrane and ensuring the normal transmission of neural signals in the body [25]. CarE, which is distributed in tissues such as the head, midgut, and fat body of insects, can hydrolyze ester bonds and amide bonds and prevent macromolecular organic compounds from acting on their targets [26,27]. GST, which is involved in the development of insect resistance, can catalyze the reaction of macromolecular organic compounds with glutathione (GSH) to produce some biological molecules [28,29]. AChE inhibits nerve impulses by hydrolyzing acetylcholine [30]. Once AChE is inhibited, acetylcholine accumulates in the insect body and triggers excitotoxicity, leading to insect death [31]. Understanding the mechanisms of action of insect enzymes against the metabolites of strain HK-1 will be helpful for further applications of this fungus.

In this study, we investigated the infection process of *A. craccivora* by *L. araneicola* HK-1. We estimated mortality, enzyme activity, and enzyme-related gene expression in aphid adults after crude extract treatment. Additionally, we isolated and purified a fungus crude extract to identify insecticidal active substances using column chromatography and gas chromatography–mass spectrometry (GC-MS). Identifying the insecticidal mechanism of strain HK-1 against aphids can provide valuable insights into the development of novel biopesticides using this fungus.

## 2. Materials and Methods

### 2.1. Fungal Culture Conditions

*L. araneicola* HK-1 was isolated from dead *A. craccivora* adults and was deposited in the China Center for Type Culture Collection (CCTCC), with the strain code CCTCC M 2022252. The identification of the strain HK-1 was published in our previous work [32].

### 2.2. Aphid Rearing

*A. craccivora* was reared on cowpea in chambers (24 ± 1 °C, 50–60% RH with a 16:8 h photoperiod). Because winged and wingless aphids may have different resistance to insecticides, wingless adult aphids that were 5 days old were selected for the bioassays to achieve accurate and stable detection results.

### 2.3. Infection Process of L. araneicola HK-1-GFP

The expression plasmid PCT-74: G418 was used in this experiment. Protoplasts of *L. araneicola* HK-1 were prepared as described [33] with some modifications. We dissolved a mixture containing 60 mg of lywallzyme (Guangdong Microbial Culture Collection Center, Guangzhou, China) and 30 mg of snailase (Biodee Biotechnology Co., Ltd., Beijing, China) using 0.7 M NaCl, to the final NaCl concentration of 30 mg mL^−1^. The mycelia were used 16 h after inoculation of the conidia, with an enzymatic digestion time of 3.5 h. The transformation and verification of the transformants were performed according to a method [33], with some modifications. Both layers of the TB3 solid medium contained G418 (800 µg mL^−1^) and ampicillin (Amp, 100 µg mL^−1^). The strain HK-1 was transformed to express eGFP as a biomarker and named *L. araneicola* HK-1-GFP.

Blastospores of the strain HK-1-GFP were obtained from 5-day-old cultures incubated in potato dextrose broth (PDB) by filtering with four layers of lens paper. A blastospore suspension (1 × 10^8^ conidia mL^−1^) in 0.02% Tween-80 in sterile water was prepared. Forty aphids were dipped into the blastospore suspension for 5 s, transferred to leafless cowpea stems whose ends were kept moisturized with wet cotton, and then placed into Petri dishes (90 mm diameter). The Petri dishes were closed and then kept in chambers (24 ± 1 °C, 50–60% RH with a 16:8 h photoperiod). Five aphids for each treatment were selected randomly and observed under an OLYMPUS BX53F fluorescence microscope (Olympus Co., Ltd., Tokyo, Japan) after treatment for 24, 36, 48, 72, 96, and 120 h. The *A. craccivora* infection process by strain HK-1 was identified via the observation of GFP expression.

### 2.4. Scanning Electron Microscopy

In this experiment, the aphids were treated as described above. After treatment for 24, 48, 72, 96, and 120 h, five aphids for each treatment were selected randomly, fixed with 2.5% glutaraldehyde for 4 h, then washed with 0.02 M phosphate-buffered saline (PBS) for three times. The aphids were placed in graded ethanol (30%, 50%, 75%, 85%, and 100%) to dehydrate for 30 min. The dehydrated samples were dipped into the ethanol–isoamyl acetate (1:1) for 10 min and maintained in absolute isoamyl acetate. After that, the aphid samples were treated with a gold spray. A scanning election microscope (TM4000Plus, Tokyo, Japan) was used to study the infection by the conidia of *A. craccivora*.

### 2.5. Measurement of Chitinase Activity

Chitinase activity was measured using the DNS method [34]. In this experiment, 1 mL of strain HK-1 blastospore suspension (1 × 10^8^ conidia mL^−1^) was inoculated in 100 mL of liquid medium (0.5% peptone; 0.05% KH_2_PO_4_; 0.05% KCl; 0.05% MgSO_4_·7H_2_O; 0.1% ZnSO_4_·7H_2_O) with a varying content of colloidal chitin (0; 0.5%; 1%; 2%; 4% *w*/*v*) and then cultured at 28 °C in a shaking incubator (180 rpm) for 3, 4, 5, 6, and 7 d. The reaction mixture containing 0.5 mL of 1% (*w*/*v*) colloidal chitin and 1 mL of enzyme solution was incubated at 50 °C for 1 h. The experiment was repeated three times. The released N-acetyl-D-glucosamine was measured spectrophotometrically at 540 nm. One unit of chitinase activity is defined as the amount of enzyme capable of digesting colloidal chitin to produce 1 µg of N-acetyl-D-glucosamine in 1 min.

### 2.6. Measurement of Extracellular Protease Activity

Extracellular protease activity was measured using the Folin method [35]. To this aim, 1 mL of blastospore suspension (1 × 10^8^ conidia mL^−1^) of strain HK-1 was inoculated in 100 mL of liquid medium (0.03% K_2_HPO_4_; 0.03% NaCl; 0.03% MgSO_4_·7H_2_O; 0.02 M phosphate buffer) containing different gelatin concentrations (0, 0.5%, 1%, 2%, 4% *w*/*v*) at 28 °C in a shaking incubator (180 rpm) for 3, 4, 5, 6, and 7 d. The reaction mixture containing 1 mL of 2% (*w*/*v*) casein solution and 1 mL of enzyme solution was incubated at 37 °C for 30 min. Then, the reaction was quenched for 20 min at 37 °C in a water bath by adding 2 mL of trichloroacetic acid (0.4 mol L^−1^). The filtrate was separated from the mixture by filtering through four layers of lens paper. Subsequently, the filtrate was transferred into a 15 mL test tube and reacted with 5 mL of Na_2_CO_3_ solution (0.4 mol L^−1^) and 1 mL of Folin phenol reagent at room temperature for 5 min. Finally, the released tyrosine was measured spectrophotometrically at 680 nm. Each measurement was performed three times. One unit of protease activity is defined as the amount of enzyme capable of digesting casein to produce 1 µg of tyrosine in 1 min.

### 2.7. Bioassay to Test the Effects of the Crude Extract in A. craccivora

The strain HK-1 was cultured in PDB in a shaking incubator at 28 °C for 7 days and then maintained in static culture conditions for 20 days in the dark. Then, the culture soup was centrifuged at 12,000 rpm for 15 min. The culture pellet was mixed with an equal volume of ethyl acetate for 12 h. The organic phase was concentrated in a rotary vacuum evaporator until the appearance of a yellow-brown precipitate. The crude extract was weighed and redissolved in acetone for the subsequent bioassays.

The leaf dipping method with some modifications was used to determine the virulence of the crude extract against *A. craccivora* adults [36]. The crude extract was diluted with acetone to obtain five different extract concentrations (7.5, 15, 30, 60, 120 mg mL^−1^); acetone was used as a control. Leafless cowpea stems, 50 mm in length, were immersed into the different extract solutions for 10 s. After acetone was volatilized naturally, the aphids were transferred on the cowpea stems moisturized with wet cotton and placed into Petri dishes (90 mm diameter). For each extract concentration, we prepared nine replicates, and each replicate (Petri dish) consisted of 20 aphids. Then the Petri dishes with the incubated aphids were placed in chambers (24 ± 1 °C, 50–60% RH with a 16:8 h photoperiod) for 24 h. After 24 h, the number of dead aphids was recorded, and aphid mortality was calculated.

### 2.8. Effect of the Crude Extract on Enzyme Activity

To estimate CarE, GST, and AChE activity, the aphids were transferred to the leafless cowpea stems treated with 24.00 mg mL^−1^ (LC_50_ value of the crude extract at 24 h) of crude extract. The treatment was consistent with that reported above. As a positive control, we treated a sample with 0.96 µg mL^−1^ (sublethal concentration (LC_30_) value of acetamiprid at 48 h) of acetamiprid, and as a negative control, we treated a sample with acetone. Then, the aphids that survived (30 mg) were homogenized using a piston in a microtube with physiological saline for 12, 24, and 48 h. Then, the homogenates were analyzed using a kit, in accordance with the manufacturer’s instructions (Nanjing Jiancheng Bioengineering Institute, Nanjing, China). The data from three replicates for each treatment were used to calculate the specific activity of each enzyme.

### 2.9. Effect of the Crude Extract on Enzyme Genes

To detect quantitative changes in enzyme expression, quantitative real-time PCR (qRT-PCR) was conducted to examine the expression of the AChE, GST, and CarE genes. Primers for the coding sequences (CDSs) of *CarE1*, *CarE2*, *CarE3*, *GST-S1*, *GST-D1*, *GST-D2*, and *AChE-X1* were designed using the Primer 5.0 software and are listed in Appendix A. They were synthesized by Sangon Biotech (Shanghai, China). Twenty living *A. craccivora* adults per group were randomly collected for RNA extraction with Trizol reagent (Sangon Biotech, Shanghai, China) after treatment with the crude extract at the LC_50_ concentration for 12, 24, and 48 h. Aphids treated with acetone were the negative control, and aphids treated with acetamiprid were the positive control. Three biological replicates for each treatment were analyzed. The first-strand cDNA was synthesized using the Primer-Script RT Reagent Kit (Takara, Dalian, China) in accordance with the manufacturer’s instruction. The synthesized cDNA was diluted to 100 µg mL^−1^ and used as a template for qRT-PCR, using the RealUniversal Color PreMix (SYBR Green) Kit (Tiangen, Beijing, China). Each qRT-PCR reaction mixture (20 µL final volume) contained 10 µL of 2× RealUniversal PreMix, 0.4 µL of 50× ROX reference dye, 1 µL of cDNA, and 0.6 µL of each primer. The qRT-PCR reaction conditions involved 15 min of initial denaturation at 95 °C, followed by 40 cycles at 95 °C for 10 s, one cycle at 60 °C for 1 min, and a melting step at 60 °C for 1 min. The gene *RPS8* of *A. craccivora* was used in a previous study as a reference [37]. *RPS8* was used to normalize the expression of the *A. craccivora* target genes. Then, the data were analyzed according to the 2^−ΔΔCT^ method [38].

### 2.10. Effect of the Crude Extract on ATPase

To estimate Na^+^-K^+^-ATPase and Ca^2+^-Mg^2+^-ATPase activities, we performed the same treatment as that described in paragraph 2.8. Then, the homogenates were processed in accordance with the instructions included in the kit from the Nanjing Jiancheng Bioengineering Institute (China). The data from three replicates for each treatment were used to calculate the specific activity of each enzyme.

### 2.11. Isolation of the Crude Extract and GC-MS Analysis

To determine its insecticidal components, the crude extract (20 g) was analyzed by column chromatography on silica gel (300–400 mesh), eluting stepwise with petroleum ether, ethyl acetate/petroleum ether (2:1), ethyl acetate/petroleum ether (1:1), dichloromethane/methanol (10:1), and methanol. Similar fractions were pooled together and concentrated on a thin-layer chromatography (TLC) plate. TLC was carried out with a mobile phase, and the results were visualized with 5% sulfuric acid in ethanol. Finally, 16 components were obtained, dissolved in acetone to a concentration of 10 mg mL^−1^, and used in the bioassay against *A. craccivora* (three replicates for each sample, with 15 aphids per Petri dish). The treatment was consistent with that described in the paragraph ‘Bioassay against *A. craccivora*’ earlier in this section.

Based on the results of the bioassays, the component No. 3 was analyzed using an Agilent 7890–7000B GC-MS (Agilent, Santa Clara, CA, USA) using the fused silica capillary column HP-5MS (30 m × 0.25 mm × 0.25 µm film thickness). Helium was used as the carrier gas at a flow rate of 1 mL min^−1^. The solvent delay was 3 min, and the loading amount was 50 µL. The temperature was increased from 50 °C to 300 °C (5 °C min^−1^), and the injector temperature was kept at 250 °C. Finally, the compounds were identified by comparing their mass spectral patterns with those reported in the National Institute of Standards and Technology (NIST) mass spectral database.

### 2.12. Data Analysis

Significance differences were compared using one-way analysis of variance (ANOVA) and Duncan’s new multiple range test (*p* < 0.05). The LC_30_ and LC_50_ values were calculated by probit analysis. All data, which were obtained from three independent experiments, are expressed as the mean ± standard error (SE) and were analyzed using SPSS 25.0 statistical software.

## 3. Results

### 3.1. Infection Process of L. araneicola HK-1-GFP

The mycelia of strain HK-1 were enzymatically digested to release the protoplasts (Appendix A). Using the PEG-mediated protoplast transformation approach, the PCT-74: G418 plasmid was successfully introduced into the protoplasts to obtain the strain HK-1-GFP (Appendix A). After treatment with HK-1-GFP, aphid infection was observed by fluorescence microscopy. After 24 h of incubation, the conidia adhered to the body surface of the aphids (Figure 1a). After 36 h, the HK-1-GFP conidia germinated to form germ tubes (Figure 1b), which further extended and developed into hyphae that grew along the epidermis on the feet of the aphids after 48 h (Figure 1c). After 72 h of incubation, foot segments were wrapped in hyphae (Figure 1d). As the degree of infection increased, the aphid abdomen became infected (Figure 1e), and the feet appeared wrapped by a large number of hyphae (Figure 1f).

### 3.2. Scanning Electron Microscopy

After inoculation with the conidia of the strain HK-1, the infection process was observed using SEM. As a control, *A. craccivora* adults were treated with 0.01% Tween-80 in water and displayed no conidia on the surface of their bodies (Figure 2a). After 24 h of treatment, HK-1 conidia were observed on the surface or grooves of the aphids, and the conidia germinated to form germ tubes (Figure 2b,c). The conidia germinated and extended to hyphae that adhered to the cuticle or appressorium, directly penetrating the cuticle of the aphids within 48 h of treatment (Figure 2d–f). As the degree of infection increased, aerial mycelia were observed on the cuticle (Figure 2g), and hyphae were twined around the mouthparts and peduncular segments of the aphids 72 h after the inoculation (Figure 2h). Many aphids died during this period. After 96 h of treatment, the feet of the aphids were entangled in many hyphae, and specialized infection pegs subsequently formed (Figure 2i,j). The mycelia grew further and produced a large number of microconidia and macroconidia using the nutrients present in the aphids. The entire body of the aphids was covered with mycelia, and the aphids died after 120 h of treatment (Figure 2k,l).

### 3.3. Measurement of Chitinase Activity

Chitinase activity gradually increased with increasing incubation time and then decreased (Table 1). On the 6th day of incubation, chitinase activity in *A. craccivora* adults, measured in 0.5% colloidal chitin liquid medium, reached a maximum of 2.48 U mL^−1^, which was significantly different from that measured without colloidal chitin (F_4,10_ = 19.983, *p* < 0.05). After the 5th day of incubation, chitinase activity in 2% colloidal chitin liquid medium was significantly higher (2.50 U mL^−1^) than in the control (2.31 U mL^−1^) (F_4,10_ = 5.99, *p* < 0.05).

### 3.4. Measurement of Extracellular Protease Activity

The extracellular protease activity of HK-1 cells increased with the increasing incubation time. After 7 days of incubation, the extracellular protease activity of strain HK-1 increased by 0, 0.5%, 1%, 2%, and 4% in cultures with a gelatin content of 4.33, 11.30, 15.65, 19.48 and 24.72 U mL^−1^, respectively. These values were 29.39-, 1.60-, 1.33-, 1.40-, and 1.37-fold higher than those measured on day 3 (‘0’: F_4,10_ = 74.641, *p* < 0.05, ‘0.5’: F_4,10_ = 47.136, *p* < 0.05, ‘1’: F_4,10_ = 38.545, *p* < 0.05, ‘2’: F_4,10_ = 7.263, *p* < 0.05, ‘4’: F_4,10_ = 13.570, *p* < 0.05) (Table 2).

### 3.5. Bioassay on A. craccivora

To evaluate the insecticidal activity of the HK-1 crude extract, *A. craccivora* adults were treated with the crude extract. The results showed that a higher aphid mortality (76.67%) was observed after treatment with the crude extract (60 mg/mL) than after acetone treatment (control; F_5,12_ = 38.681, *p* < 0.05; Appendix A), and the LC_50_ value was 24.00 mg mL. Even if the mortality rate of the aphids was 10.00% after 24 h of acetone treatment, there was a significant difference between the two treatments.

### 3.6. Effect of the Crude Extract on Enzyme Activity

To evaluate the effect of the crude extract on enzyme activity in *A. craccivora* adults, acetamiprid-treated aphids were used as a positive control (Appendix A). The results showed that a higher aphid mortality was observed after acetamiprid treatment than after acetone treatment (control; F_5,12_ = 161.289, *p* < 0.05). After 48 h of treatment, the LC_50_ value and the LC_30_ value were 1.84, 0.96 µg mL^−1^, respectively.

CarE activity in *A. craccivora* adults treated with the crude extract was determined (Figure 3a). The crude extract and acetamiprid caused a significant reduction (‘crude extract’: F_1,4_ = 76.746, *p* < 0.05, ‘acetamiprid’: F_1,4_ = 109.184, *p* < 0.05) in the activity of CarE by 50.09% and 49.46%, respectively after 24 h. In addition, GST activity in *A. craccivora* adults treated with the crude extract was significantly higher (‘12 h’: F_1,4_ = 23.810, *p* < 0.05, ‘24 h’: F_1,4_ = 89.285, *p* < 0.05, ‘48 h’: F_1,4_ = 53.079, *p* < 0.05) than in the control at the three treatment times (Figure 3b). At 12, 24, and 48 h, GST activity in the crude extract group was 1.73-, 1.69- and 1.77-fold higher than in the control group, respectively. AChE activity in adult *A. craccivora* treated with the crude extract for 24 h was significantly higher—1.31-fold—than in the control group (F_1,4_ = 7.242, *p* < 0.05; Figure 3c).

### 3.7. Effect of the Crude Extract on Enzyme Genes

The effect of the crude extract on the transcription levels of seven enzyme genes (*CarE1*, *CarE2*, *CarE3*, *GST-S1*, *GST-D1*, and *GST-D2*, and *AChE-X1*) in *A. craccivora* adults was determined (Figure 4). The expression of the *CarE1* gene was upregulated significantly in the crude extract and acetamiprid groups after treatment for 48 h (‘crude extract’: F_1,4_ = 920.459, *p* < 0.05, ‘acetamiprid’, F_1,4_ = 22.612, *p* < 0.05) and was 5.05-, 6.32-fold higher than in the control group, respectively (Figure 4a). As shown in Figure 4b, *CarE2* expression in the crude extract group was 49.37% of that of the control and was significantly downregulated (F_1,4_ = 18.053, *p* < 0.05) after 24 h of treatment. The expression of the *CarE3* gene in the crude extract group was significantly lower than that in the control group (58.27% inhibition) after 24 h of treatment (F_1,4_ = 18.866, *p* < 0.05; Figure 4c).

The expression of the *GST-S1* gene was significantly upregulated in the crude extract and acetamiprid groups after treatment for 12 h (‘crude extract’: F_1,4_ = 258.389, ‘acetamiprid’, F_1,4_ = 112.857, *p* < 0.05) and was 6.32 and 1.92 times higher than that in the control group, respectively (Figure 4d). The *GST-D1* gene expression in the crude extract group and acetamiprid group was significantly upregulated and was 2.04- and 1.93-fold higher than in the control group after treatment for 12 h, respectively (‘crude extract’: F_1,4_ = 38.154, ‘acetamiprid’, F_1,4_ = 13.236, *p* < 0.05). The *GST-D1* gene expression in the crude group and acetamiprid group was 8.21- and 3.70-fold higher than in the control group, showing significant upregulation after 48 h (‘crude extract’: F_1,4_ = 2147.704, ‘acetamiprid’, F_1,4_ = 43.167, *p* < 0.05), respectively (Figure 4e). The *GST-D2* gene expression in the crude extract group was significantly upregulated and was 4.58- and 2.02-fold higher than in the control group at 12 h and 24 h (‘12 h’: F_1,4_ = 16.012, ‘24 h’, F_1,4_ = 17.298, *p* < 0.05), respectively (Figure 4f).

The *AChE-X1* gene expression in the crude extract group was significantly upregulated (‘12 h’: F_1,4_ = 7.643, ‘48 h’, F_1,4_ = 13.480, *p* < 0.05) and was 1.82- and 1.50-fold higher than in the control group after treatment for 12 h and 48 h, respectively (Figure 4g).

### 3.8. Effect of the Crude Extract on ATPase

The ATPase activity in *A. craccivora* adults was evaluated after treatment with the crude extract (Figure 5). The Na^+^-K^+^-ATPase activity in treated *A. craccivora* was 90.13% of that in the control group after treatment for 12 h, indicating a significant inhibition in the crude extract group (Figure 5a, F_1,4_ = 24.060, *p* < 0.05). But it was notably induced after 24 h and 48 h, with a 1.23- and a 1.24-fold increase with respect to the of control, respectively (Figure 5a, ‘24 h’: F_1,4_ = 12.55, *p* < 0.05, ‘48 h’: F_1,4_ = 54.861, *p* < 0.05). The Ca^2+^-Mg^2+^-ATPase activity in the crude extract group and the acetamiprid group was significantly induced after 48 h by 1.27- and 1.15-fold with respect to that in the control, respectively (Figure 5b, ‘crude extract’: F_1,4_ = 23.426, *p* < 0.05, ‘acetamiprid’: F_1,4_ = 33.601, *p* < 0.05).

### 3.9. Isolation of the Crude Extract and GC-MS Analysis

Using silica column chromatography, 16 components were obtained and tested in the bioassay. The best insecticidal components were No. 3 and No. 12, and the average mortality they caused was 64.44% and 44.44%, respectively (Appendix A). In total, 124 different compounds were identified in component No. 3, accounting for 79.78% of the constituents of component No. 3 (Appendix A). The primary constituents were aromatic compounds (14.67%), alcohols (11.94%), alkanes (8.99%), ketones (6.17%), amines (6.03%), acids (5.23%), phenols (2.18%), aldehydes (3.55%), and esters (3.3%).

## 4. Discussion

*Leacnicillium* is an important entomopathogenic fungus with a wide geographical distribution and a broad host range [39]. However, there are few studies on the infection and insecticidal mechanism of *Lecanicillium* in *A. craccivora*. In this study, we showed that the infection of *A. craccivora* by strain HK-1 and the resulting metabolites were lethal to the aphids.

The process of *A. craccivora* infection by strain HK-1 included conidial attachment and germination, hyphae development, invasion, and *A. craccivora* death. However, the processes through which fungi infect different insects vary. Gao et al. reported that the conidia of *L. lecanii* formed appressoria directly, and the hyphae formed penetration pegs or produced copious mucilage to infect *Phenacoccus fraxinus* [40]. In contrast, penetration structures and mucilage were not observed during the infection by *V. lecanii* DAOM 198499 of *Macrosiphum euphorbiae* [41]. Specific observations were made in the present study. The appressoria formed from the ends of the germ tubes, rather than directly from the conidia (Figure 2f). In terms of energy utilization, conidial development using nutrients is more economic than invasion through the integument [41]. Furthermore, the cuticle is concave at the appressorium site (Figure 2f), suggesting that the appressorium produces hydrolases, including chitinases and proteases [42]. Chitinases and proteases, which are important virulence factors for entomopathogenic fungi, can hydrolyze the insect cuticle during infection [43]. Additionally, the waxy cuticle distributed on the insect body surface plays a role in the defense against fungal infections. The waxy layer on the ventral area of *Ceroplastes japonicus* was thinner than on the dorsal side, and the conidia of *L. lecanii* adhered to the ventral plate, primarily at the base of the mouthparts and legs [20]. In contrast, despite the fact that the body surface of *P. fraxinus* is covered with various defense substances, the hyphae of *L. lecanii* could easily pass through the waxy filaments and the wet waxy agglomerations to successfully infect *P. fraxinus* [40]. In our study, we demonstrated that the strain HK-1 could generate chitinase and protease, which played a major role in the infection process of *A. craccivora*. Furthermore, we also observed that the body surface of *A. craccivora* was covered with a wax layer, which was thinner in the dorsum and feet than in other areas (Figure 2d–i). The results of the infection process with the strain HK-1 indicated that the conidia easily attached to the feet and dorsum of the insects.

During the infection of insects, entomopathogenic fungi secrete secondary metabolites to neutralize the host immune mechanisms, allowing them to overcome the immune defense system and accelerate host death [44,45,46]. Many studies reported that the toxins secreted by entomopathogenic fungi have strong insecticidal activity [47]. For example, the crude ethyl acetate extracts of six entomopathogenic fungi (*Hirsutella*) showed insecticidal toxicity against adult females of *Tetranychus urticae*, and the highest and the lowest LC_50_ values were 44,197.85 and 1555.46 mg L^−1^, respectively [47]. Furthermore, the crude acetone extract of *L. attenuatum* JEF-145 cultured on a solid rice medium showed insecticidal activity against both *Aedes albopictus* and *Plutella xylostella* [48]. In this study, the crude ethyl acetate extract of strain HK-1 showed contact activity against *A. craccivora* adults using the leaf-dipping method, with an LC_50_ value of 24.00 mg mL^−1^ after 24 h of treatment (Appendix A).

Insects produce enzymes, including CarE and GST, to metabolize harmful exogenous substances [25]. AChE is also induced to degrade acetylcholine, to ensure the normal transmission of neural signals [25]. CarE structure includes an *α/β* folded domain, an acyl binding domain, and a catalytic triad, resulting in the ability to hydrolyze ester bonds [26]. As an enzyme, an increase in the activity of GST is related to an increase in insect resistance [49]. AChE exists in the presynaptic membrane, postsynaptic membrane, and synaptic cleft and can hydrolyze acetylcholine to maintain the normal transmission of nerve impulses [50]. AChE activity is inhibited by xenobiotics, including *Triadica sebifera*, *Basilicum ocimum*, and *Origanum marjorana* [51,52], which results in the accumulation of acetylcholine in the insect body, triggering excitotoxicity [53]. In this study, acetamiprid inhibited aphid CarE activity, which is consistent with previous reports [54]. Following treatment with the crude extract, CarE activity was significantly inhibited in *A. craccivora* after 24 h (Figure 3a), and the expression of the enzyme genes *CarE2* and *CarE3* was significantly downregulated (Figure 4b,c). It is suggested that the crude extract, with its effects on CarE, prevented the normal metabolism in *A. craccivora*. The upregulated expression of the GST genes maintained a high GST activity, indicating that the aphids continuously tried to alleviate the toxic effects of the crude extract (Figure 3b). Additionally, AChE activity significantly enhanced the hydrolytic capacity of *A. craccivora* for acetylcholine and relieved nerve impulses after 24 h. In summary, the results indicated that the crude extract disrupted the enzyme system of *A. craccivora*, impacting on the aphid physiological activities.

Additionally, Na^+^-K^+^-ATPase is abundantly present on the cytoplasmic membrane of excitatory cells such as neuronal synapses and plays a vital role in maintaining the balance between Na^+^ and K^+^, which is crucial for the conduction of nerve impulses [55]. Ma studied the changes in *Moina macrocopa* treated with deltamethrin and found that after treated 24 h, Na^+^-K^+^-ATPase activity was reduced significantly [56]. Leng et al. found that deltamethrin inhibited Na^+^-K^+^-ATPase in housefly brain synaptosomes at concentrations ranging from 10^−9^ to 10^−6^. These studies indicate that pesticides produce inhibitory effect on Na^+^-K^+^-ATPase in insects’ synaptosomes [57]. The results presented in our paper also showed that the activity of Na^+^-K^+^-ATPase in *A. craccivora* bodies was significantly inhibited after 12 h of treatment with the crude extract (Figure 5a), consistent with the effects of other pesticides. Ca^2+^-Mg^2+^-ATPase is also involved in ion and protein trafficking, the maintenance of cellular homeostasis, and cell signaling [58]. When Ca^2+^-Mg^2+^-ATPase is inhibited, calcium homeostasis is disturbed, leading to metabolic dysfunctions in insects [59,60]. In the present study, Ca^2+^-Mg^2+^-ATPase activity in *A. craccivora* was induced by the crude extract 48 h after treatment (Figure 5b). These changes might contribute to maintaining the relative stability of the neuronal membrane potential, as we observed that even after 24 h of treatment with the crude extract, *A. craccivora* could still crawl. Further studies need to be conducted to verify this hypothesis.

In this study, we also identified the components of the crude extract that may be related to insecticidal activity. Previous studies reported that p-cymene [61,62,63], hymecromone [64,65,66], 9,12-octadecadienoic acid (Z, Z)-methyl ester [67], and 9,12-octadecadienoic acid (Z, Z) [68] have insecticidal effects on insects, and these compounds were also found in our study (component no. 3). The proportion of these compounds in component No. 3 ranged from 0.09% to 3.91% (Appendix A). These compounds may be related to the insecticidal activity we observed against aphids, although further studies are required.

## 5. Conclusions

In summary, HK-1 strain had the capability to induce death in the aphid *A. craccivora*. Treatment with the crude extract of HK-1 led to alterations in enzyme activity (protease, chitinase, and ATPase) in the aphids as well in the expression of enzyme genes, predominantly inducing their upregulation. Through gas chromatography–mass spectrometry (GC-MS) analysis, we identified substances that may be linked to the insecticidal effects of the HK-1 strain on aphids. However, further experiments are required to isolate these compounds and confirm their activities. Our data provide a basis for further exploration of the potential use of strain HK-1 as a biocontrol agent against *A. craccivora* and to enhance the development of entomopathogenic fungi to control aphids and other harmful insects in agriculture.

## Figures and Tables

**Figure 1 insects-14-00860-f001:**
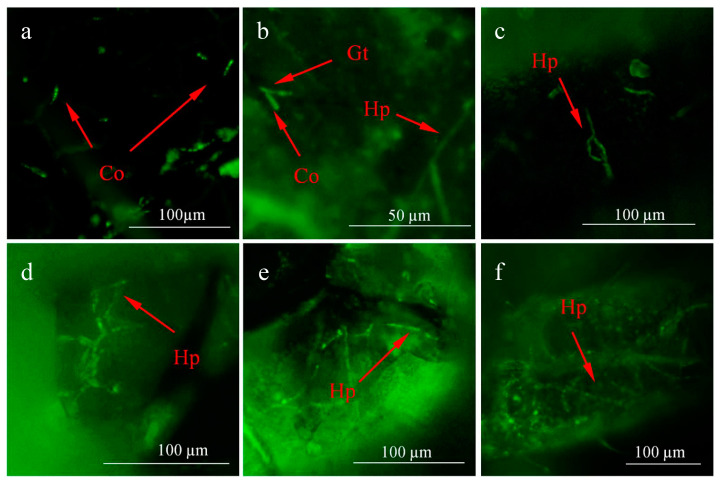
Infection process of the strain HK-1-GFP in *A. craccivora*. (**a**) After 24 h of infection with HK-1-GFP, conidia (Co) were visible on the surface of aphids. (**b**) After 36 h of infection, germination formed germ tube (Gt) and hyphae (Hp). (**c**) After 48 of infection, the hyphae (Hp) were wrapped on the aphid foot. (**d**) After 72 h of infection, foot segments were wrapped by hyphae (Hp). (**e**) After 96 h of infection, the abdomen was infected by hyphae (Hp). (**f**) After 120 h of infection, the feet were wrapped by a large number of hyphae (Hp).

**Figure 2 insects-14-00860-f002:**
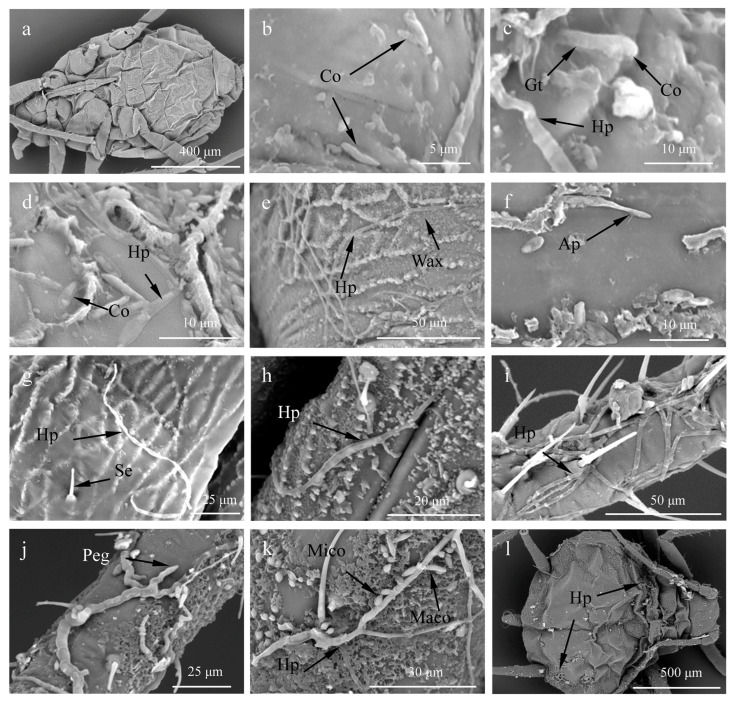
Scanning electron micrographs of the invasion of strain HK-1 on the surface of *A. craccivora*. (**a**) *A. craccivora* adults treated with 0.01% Tween-80 in water. (**b**) Few conidia (Co) attached onto the body surface. (**c**) Conidia (Co) germination to form a germ tube (Gt). (**d**) Conidia (Co) and hyphae (Hp) on the back of *A. craccivora*. (**e**) Hyphae (Hp) interweaved to form a network structure. (**f**) Appressorium (Ap). (**g**) Hyphae (Hp). (**h**) Hyphae (Hp) infecting the mouthparts of *A. craccivora*. (**i**) Hyphae (Hp) twining around the foot of *A. craccivora*. (**j**) Penetration (Peg). (**k**) Hyphae (Hp) produced microconidia (Mico) and macroconidia (Maco). (**l**) *A. craccivora* infected by strain HK-1 on the 5th day of treatment.

**Figure 3 insects-14-00860-f003:**
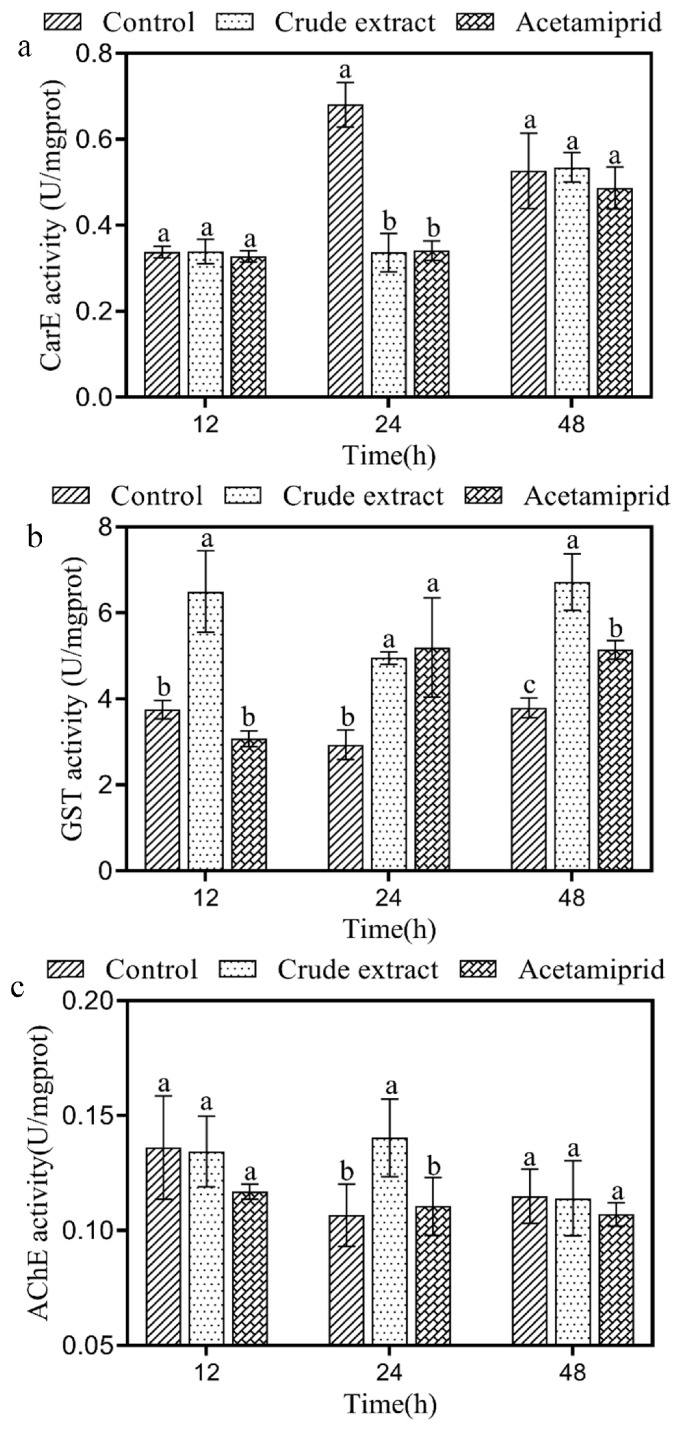
Effect of the crude extract on CarE (**a**), GST (**b**), and AChE (**c**) enzyme activity in *A. craccivora*. Error bars indicate SE across three replicates. Different small letters indicate significant differences between different treatments at the same treatment time (*p* < 0.05).

**Figure 4 insects-14-00860-f004:**
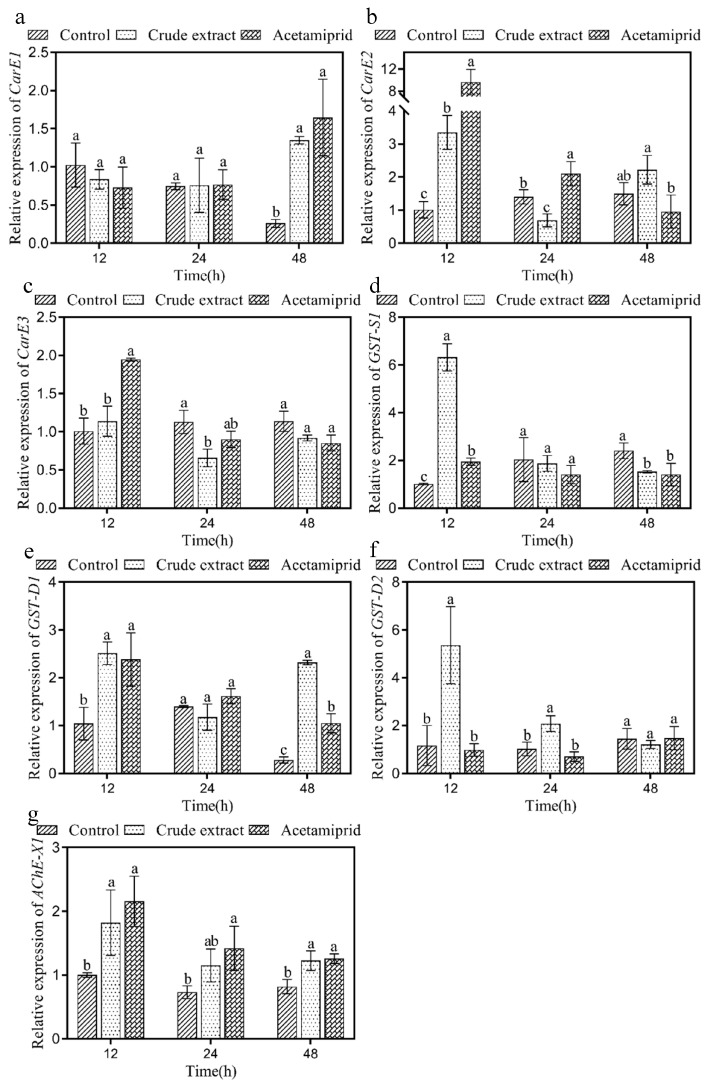
Effect of the crude extract on *CarE1* (**a**), *CarE2* (**b**), *CarE3* (**c**), *GST-S1* (**d**), *GST-D1* (**e**), *GST-D2* (**f**), and *AChE-X1* (**g**) expression in *A. craccivora*. Error bars indicate SE across three replicates. Different small letters indicate significant differences between different treatments at the same time point (*p* < 0.05).

**Figure 5 insects-14-00860-f005:**
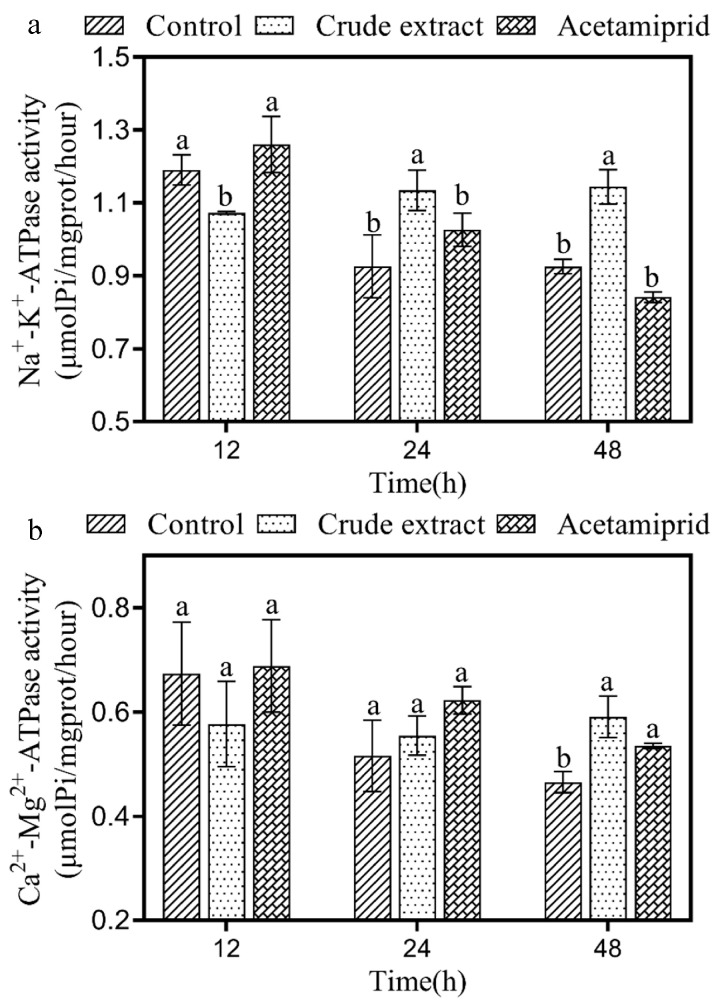
Effect of the crude extract on Na^+^-K^+^-ATPase (**a**) and Ca^2+^-Mg^2+^-ATPase (**b**) activity in *A. craccivora*. Error bars indicate SE across three replicates. Different small letters indicate significant differences between different treatments at the same time point (*p* < 0.05).

**Table 1 insects-14-00860-t001:** Chitinase activity of strain HK-1 in culture media with different colloidal chitin contents for 3–7 days.

Colloidal Chitin (%)	Chitinase Activity Mean ± SE (U mL^−1^)				
	3 d	4 d	5 d	6 d	7 d
0	2.13 ± 0.05 Da	2.22 ± 0.05 Cc	2.31 ± 0.04 ABb	2.37 ± 0.05 Ab	2.27 ± 0.04 BCa
0.5	2.15 ± 0.06 Ca	2.37 ± 0.05 Ba	2.32 ± 0.04 Bb	2.48 ± 0.06 Aa	2.31 ± 0.01 Ba
1	2.17 ± 0.06 Ca	2.32 ± 0.08 ABab	2.41 ± 0.11 Aab	2.25 ± 0.09 Cc	2.24 ± 0.06 Ca
2	2.12 ± 0.03 Ca	2.26 ± 0.03 Bbc	2.5 ± 0.05 Aa	2.17 ± 0.04 Ccd	2.14 ± 0.04 Cb
4	2.08 ± 0.06 Ca	2.23 ± 0.03 Bc	2.47 ± 0.02 Aa	2.14 ± 0.03 Cd	2.06 ± 0.05 Cc

Different capital letters in the table indicate significant differences (*p* < 0.05) in chitinase activity for different culture periods in the presence of the same colloidal chitin content. Different small letters in the table indicate significant differences (*p* < 0.05) in chitinase activity for media with different colloidal chitin contents on the same day.

**Table 2 insects-14-00860-t002:** Extracellular protease activity of strain HK-1 in culture medis with different gelatin contents.

Gelatin (%)	Extracellular Protease Activity Mean ± SE (U mL^−1^)				
	3 d	4 d	5 d	6 d	7 d
0	0.15 ± 0.13 Bd	0.21 ± 0.02 Be	0.10 ± 0.10 Be	0.12 ± 0.13 Be	4.33 ± 0.81 Ae
0.5	7.05 ± 0.49 Cc	5.90 ± 0.45 Cd	4.40 ± 0.70 Dd	9.69 ± 0.99 Bd	11.30 ± 0.76 Ad
1	11.78 ± 0.94 Cb	11.06 ± 0.66 Cc	11.11 ± 0.17 Cc	13.94 ± 0.45 Bc	15.65 ± 0.17 Ac
2	13.88 ± 2.76 Cb	16.12 ± 0.53 BCb	15.85 ± 0.86 BCb	17.97 ± 0.17 ABb	19.48 ± 0.91 Ab
4	18.03 ± 0.23 Da	18.52 ± 0.89 CDa	20.46 ± 1.90 BCa	21.48 ± 1.77 Ba	24.72 ± 0.56 Aa

Different capital letters in the table indicate significant differences (*p* < 0.05) in extracellular protease activity for different culture periods in the presence of the same gelatin content. Different small letters in the table indicate significant differences (*p* < 0.05) in extracellular protease activity for media with different gelatin contents on the same day.

## Data Availability

The data that support the findings of this study are available from the corresponding author upon reasonable request.

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
