# Peer review of "Insecticidal Effect of the Entomopathogenic Fungus Lecanicillium araneicola HK-1 in Aphis craccivora (Hemiptera: Aphididae)"

_insects, 2023, doi:10.3390/insects14110860_

Round 1

Reviewer 1 Report

Comments and Suggestions for Authors

The manuscript from Liu et al investigates the infection process of Lecanicillium araneicola HK-1 to Aphis craccivora, an important crop pest.

In its present form, the manuscript has not the potential to be published. The manuscript’s most significantly flaw lies in its writing (I think most of the text should be rewrite slightly), characterized by overly verbose sentences. In addition, at the first sight, the insecticide background of the manuscript is not well described. References in the manuscript are not always relevant.

Abstract

I suggest the authors get some external assistance to turn the text into more appropriate and digestible content.

Introduction

I don’t think that the first reference [1] is relevant at this position ! The cited manuscript refers to the genetic structure of Aphis Craccivora from Thailand, in particular they examined genetic structure and genetic relationship. The same also with reference [3] ! Please, check references in the manuscript to be sure that they are appropriate.

Line 44: add a reference at the end of “…. virus,”

Line 47-49: rewrite the sentence

Line 62-63 : ad reference at the end of “… in 1861,”

Line 64-66: rewrite the sentence

I am sorry, but I stopped to read the introduction which need appropriate external peer review.

Materials and methods

Line 98 : what is the meaning of “collected from the natural infected A. craccivora” ? How this infestation was determined ?

Line 103 : explain :”The wingless adult aphids with the same size and age were selected for bioassays”. Which size ? how old ? Why wingless ?

Line : be clear “ratio 2:1” ????

The number of aphids used for the experiments is confused. Be clear and specify the number of aphids used and the number of repeated experiments.

Results

Bioassay against A. Craccivora : line 315 : The results showed that at higher aphids mortality ??? be clear. Which value ?

Line 322 : this part refers to acetamiprid which is a neonicotinoid insecticide. I don’t understand why authors used this compound. Nothing in introduction and materials and methods ???? What is the link between acetamiprid and detoxification enzyme used in the present study ?

Conclusion

The conclusion is not necessarily in lien with the observed results and it remains difficult to follow.

Comments on the Quality of English Language

Need extensive peer review

Author Response

Dear Reviewer1

Thank you for your letter and comments concerning our manuscript entitled ‘Infection and insecticidal mechanism of Lecanicillium araneicola HK-1, Manuscript ID: insects-2639039. We have made a careful revision on the original manuscript and tried to avoid any mistake.

Thank you for your consideration. I look forward to hearing from you.

Sincerely yours,

Dr. Pengfei Jin

School of Plant Protection, Hainan University, No. 58 Renmin Avenue, Haikou, Hainan, 570228, P.R. China

Email: jinpengfei@hainanu.edu.cn

Reviewer1

Question 1: The manuscript from Liu et al investigates the infection process of Lecanicillium araneicola HK-1 to Aphis craccivora, an important crop pest. In its present form, the manuscript has not the potential to be published. The manuscript’s most significantly flaw lies in its writing (I think most of the text should be rewrite slightly), characterized by overly verbose sentences. In addition, at the first sight, the insecticide background of the manuscript is not well described. References in the manuscript are not always relevant.

Response: Thank you for your suggestion. We added the information about insecticide background and rewrote the sentence in more detail (line 45-53: Insecticide application remains the primary method of aphid control in field crops including the pyrethroid, organophosphorus and carbamates etc. [10]. However, the frequent use and abuse of chemical pesticides results in great resistance to a wide range of insects, environmental pollution, soil deterioration, and high pesticide residues world-wide [11, 12]. Therefore, it is imperative to develop safe, effective, and sustainable biologi-cal methods for aphid control. The application of entomopathogenic fungi is a promising alternative approach for aphid control because of their low cost, environmental friendly, innocuous nature to non-target organisms, and wide range of raw material sources [13].).

We have checked all references, deleted any non-relevant ones, and replaced them with relevant references.

Question 2: I suggest the authors get some external assistance to turn the text into more appropriate and digestible content.

Response: We appreciate very much for the reviewer’s valuable comments. We have fully revised the manuscript of language issues that make the manuscript turn into more appropriate and digestible content.

Question 3: I don’t think that the first reference [1] is relevant at this position! The cited manuscript refers to the genetic structure of Aphis Craccivora from Thailand, in particular they examined genetic structure and genetic relationship. The same also with reference [3] ! Please, check references in the manuscript to be sure that they are appropriate.

Response: Thanks, We thoroughly reviewed the manuscript, checked all references, deleted any non-relevant ones, and replaced them with relevant references. We have replaced the reference [1] with Dixon A.F.G.; Aphid ecology: an optimization approach. 2nd edition. London: Chapman and Hall;1998. and reference [3] replanced with Choudhary A. L.; Hussain A.; Choudhary M. D.; Samota R.; Jat S. Bioefficacy of newer insecticides against aphid, Aphis craccivora Koch on cowpea. J Pharmacogn Phytochem 2017, 6: 1788-1792.

Question 4: Line 44: add a reference at the end of “…. virus,”

Response: Thanks for your suggestion, and we have added relevant reference [6] (Blackman RL, Eastop VF: Aphids on the world's crops: an identification and information guide. Chichister: John Wiley & Sons; 2000.)

Question 5: Line 47-49: rewrite the sentence

Response: We have rewritten the sentence(P2.Line 45-53: Insecticide application remains the primary method of aphid control in field crops including the pyrethroid, organophosphorus and carbamates etc. [10]. However, the frequent use and abuse of chemical pesticides results in great resistance to a wide range of insects, environmental pollution, soil deterioration, and high pesticide residues world-wide [11, 12]. Therefore, it is imperative to develop safe, effective, and sustainable biologi-cal methods for aphid control. The application of entomopathogenic fungi is a promising alternative approach for aphid control because of their low cost, environmental friendli-ness, innocuous nature to non-target organisms, and wide range of raw material sources [13].).

Question 6: Line 62-63 : ad reference at the end of “… in 1861,”

Response: Thanks, we have added a relevant reference (Blackman RL, Eastop VF: Aphids on the world's crops: an identification and information guide. Chichister: John Wiley & Sons; 2000.)

Question 6: Line 64-66: rewrite the sentence

Response: We have rewritten the sentence. (P2. Line 62-64: The fungus Lecanicillium, originally isolated from Lecanii coffeae in Ceylon by Nivter in 1861, is an important entomopathogenic fungus of the Hypocreales which is used as a biocontrol agent for pests, including thrips, aphids, and whiteflies[20].).

Question 7: I am sorry, but I stopped to read the introduction which need appropriate external peer review.

 Response: Thank you very much for your advice. We have fully revised the manuscript of language issues that make the manuscript turn into more appropriate and digestible content.

Materials and methods

Question8: Line 98 : what is the meaning of “collected from the natural infected A. craccivora” ? How this infestation was determined ?

Response: We rewrote the sentence (Line 98-100: L. araneicola HK-1 was isolated from dead A. craccivora adults, and was deposited in the China Center for Type Culture Collection (CCTCC) as the strain code CCTCC M 2022252.). After isolated it, we identified this species combined with morphological identification and physical and chemical properties, this was reported in our previous work (Liu et al. 2022, In Chinese).

Question9: Line 103: explain :”The wingless adult aphids with the same size and age were selected for bioassays”. Which size ? how old ? Why wingless ?

Response: We appreciate very much for the reviewer’s valuable comments. Since winged and wingless aphids may have different resistance to insecticides, and winged aphids are not suitable for bioassays due to their ability to fly, to achieve accurate and stable detection results, we selected wingless adult aphids (5 days old).To make it clearer, we rewrite the sentence as “Because winged and wingless aphids may have different resistance to the insecticide, wingless adult aphids that were 5 days old were selected for bioassays to achieve accurate and stable detection results.”(Line 121)

Question10: Line 109 : be clear “ratio 2:1” ????

Response: We have rewritten this sentence with more detailed description (P3. Line 107-111: The protoplasts of L. araneicola HK-1 were prepared as described in paper[32] with some modifications: Dissolve the mixture [60 mg of lywallzyme (Guangdong Microbial Culture Collection Center, China) and 30 mg of snailase (Biodee Biotechnology Co. Ltd, China)] with 0.7M of NaCl at concentrations of 30 mg mL-1.).

Question11: The number of aphids used for the experiments is confused. Be clear and specify the number of aphids used and the number of repeated experiments.

Response: We have rewritten the sentence with “For each concentration gradient, there were nine replication and each replication (petri dishes) consisted of 20 aphids.” (Line 205)

Question12: Bioassay against A. Craccivora : line 315 : The results showed that at higher aphids mortality ??? be clear. Which value ?

Response: We appreciate very much for the reviewer’s valuable comments. We have rewritten the sentence (P8.Line 318-323: To evaluate the insecticidal activity of the HK-1 crude extract, A. craccivora adults were treated with the crude extract. The results showed that a higher aphid’s mortality (76.67%) was observed in crude extract (60mg/mL) treatment compared to acetone (control treatment) (F5,12 = 38.681, P < 0.05; Supplementary Table S2), and the LC50 value was 24.00 mg mL. However, the mortality rate of aphids was 10.00% obtained 24 h after the acetone treatment, there was significant difference between treatments.

Question13: Line 322: this part refers to acetamiprid which is a neonicotinoid insecticide. I don’t understand why authors used this compound. Nothing in introduction and materials and methods ???? What is the link between acetamiprid and detoxification enzyme used in the present study?

Response: We appreciate very much for the reviewer’s valuable comments. Acetamiprid has been widely used in agricultural, domestic, and public health activities over the past decade to replace more dangerous insecticides such as organophosphorus, carbamate, and pyrethroid. In addition, acetamiprid exposure interrupts nerve transmission, alters membrane potential and ultimately results in neuronal hyper excitation, paralysis and death. (Phogat A.; Singh J.; Kumar V.; Malik V. Toxicity of the acetamiprid insecticide for mammals: A review. Environ Chem Lett 2022, 1-26.). As these results, we use acetamiprid to compare with the metabolites produced by the strain HK-1, which is a positive control.

Conclusion

Question14: The conclusion is not necessarily in lien with the observed results and it remains difficult to follow.

Response: We appreciate very much for the reviewer’s valuable comments. We have rewritten the whole conclusion with “In summary, HK-1 strain has the capability to induced mortality in aphids A. craccivora. Treatment with the crude extract of HK-1 led to alterations in the enzymes (protease, chitinase, and ATPase) of the aphids as well as a considerable influence on the expression of genes associated with detoxification enzymes, predominantly inducing upregulation. Through gas chromatography-mass spectrometry (GC-MS) analysis, we identified substances that may be linked to the insecticidal effects on aphids. However, further experiments are required to isolate and confirm these specific active compounds. Our data provide a basis for further exploration of the potential use of strain HK-1 as a biocontrol agent against A. craccivora and to enhance the development of entomopathogenic fungi to control aphids and other harmful insects in agriculture.” (P14.Line 492-501).

Reviewer 2 Report

Comments and Suggestions for Authors

In this study, infect process, biochemical  characteristics, and potential insecticidal compounds of an entomopathogenic fungus strain Lecanicillium araneicola HK-1 were conducted using the important crop insect, Aphis craccivora, which provides information for the development of novel biopesticides. However, there are several major concerns should be overcome before the consideration of manuscript publish.

1. In this study, insecticidal mechanism of strain KH-1 to Aphis craccivora was explored. However, there appeared to be critical flaws in the experiment designs. It is a good result that No.3 component obtained by silica column chromatography exhibited a higher insecticidal activity against aphids than the ethyl acetate crude extract, and the compounds of No.3 component were identified by GC-MS. But we cannot say "The Gas Chromatography Mass Spectrometry (GC-MS) analysis suggested that the insecticidal compounds were p-Cymene, Hymecromone, 9, 12-Octadecadienoic acid (Z, Z)-, methyl ester, and 9, 12-Octadecadienoic acid (Z, Z)." without any experimental verification.
2. According to the results, we cannot find any evidence to support "the crude extract has neurotoxic effects for the A. craccivora". Any neurotoxic lethal phenotype found? Any neurophysiology results? Effect on Na+-K+-ATPase is not equal to "neurotoxic effect". Na+-K+-ATPase is also enriched in the insect excretory system as well as in the nervous system.
3. Acetylcholinesterase (AChE) is not a detoxification enzyme for xenobiotic substances, but a mode of action of some insecticides such as organophosphorus compounds.
4. In the biochemical tests, the crude extract of the LC50 concentration at 24h was used for the treatment. The survived aphids after 12, 24 and 48 h treatment were sampled. What about the mortality after 48 h treatment? If the mortality is very high, the biochemical results of 48 h treatment is not good for analyzing the biochemical effect of fungus extract on aphids. Because the survived insect under the high lethal pressure might be high tolerant, be dying with low physiological status, or with low fitness.
5. The title need to be changed. we cannot find any strong evidence for the "insecticidal mechanism".
6. Abstract & "5. Conclusions": must be rewritten. Some indications without any evidence cannot be in these two sections.
7. 2.1: The origin and identification information, or the reference of the HK-1 strain should be provided.
8. 2.5-2.6: Replication settings should be described.
9. Provide the probit analysis software information.
10. Table S3: there is only one data point below 50% mortality. This toxicity data is not good for the calculation of LC values.

Author Response

Dear Reviewer2

Thank you for your letter and comments concerning our manuscript entitled ‘Infection and insecticidal mechanism of Lecanicillium araneicola HK-1, Manuscript ID: insects-2639039. We have made a careful revision on the original manuscript and tried to avoid any mistake.

Thank you for your consideration. I look forward to hearing from you.

Sincerely yours,

Dr. Pengfei Jin

School of Plant Protection, Hainan University, No. 58 Renmin Avenue, Haikou, Hainan, 570228, P.R. China

Email: jinpengfei@hainanu.edu.cn

Reviewer2

Question 1. In this study, insecticidal mechanism of strain KH-1 to Aphis craccivora was explored. However, there appeared to be critical flaws in the experiment designs. It is a good result that No.3 component obtained by silica column chromatography exhibited a higher insecticidal activity against aphids than the ethyl acetate crude extract, and the compounds of No.3 component were identified by GC-MS. But we cannot say "The Gas Chromatography Mass Spectrometry (GC-MS) analysis suggested that the insecticidal compounds were p-Cymene, Hymecromone, 9, 12-Octadecadienoic acid (Z, Z)-, methyl ester, and 9, 12-Octadecadienoic acid (Z, Z)." without any experimental verification.
Response: We appreciate very much for the reviewer’s valuable comments.

Our expression was incorrect, we rewrote the sentence “In this study, we also identified some components in the crude extract which may be related to the insecticidal ability. Previous studies have reported that p-Cymene [59-61], Hymecromone [62-64], 9,12-Octadecadienoic acid (Z,Z)-methyl ester [65], and 9,12-Octadecadienoic acid (Z,Z) [66] have insecticidal effects on insects, and these compounds were also found in our study (in the No. 3 component). The proportion of these compounds in the No. 3 component ranged from 0.09% to 3.91% (Supplementary Table S5). These components may be related to the insecticidal activity on aphids, although further research is needed (Line 548-555).

Question 2. According to the results, we cannot find any evidence to support "the crude extract has neurotoxic effects for the A. craccivora". Any neurotoxic lethal phenotype found? Any neurophysiology results? Effect on Na+-K+-ATPase is not equal to "neurotoxic effect". Na+-K+-ATPase is also enriched in the insect excretory system as well as in the nervous system.
Response: We appreciate very much for the reviewer’s valuable comments.

During our observation and counting process, we did indeed discover the strange movements and behaviors of aphids, and combined with the Na+- K+- ATPase enzyme we tested, additional, Qin et al showed that Na+, K+-ATPase is the ion pump of Na+ and K+ of epicyte of insect, which play a crucial role in keeping the ionic equilibrium and nerve impulse of insect body. When the activity of Na+, K+-ATPas was inhibited, the ion migration of epicyte would be blocked, leading to metabolic disturbance and nerve agent poisoning of organism. we speculated that the crude extract is related to neurotoxicity (54-57). For these reasons, we speculate that the crude extract may cause toxicity to aphid nerves.

Lees G.J. Inhibition of sodium-potassium-ATPase: a pontentially ubiquitous mechanism contributing to central nervous system neuropathology. Brain Res Rev 1991, 16, 283-300, https://doi.org/10.1016/0165-0173(91)90011-V.

Qin W.; Huang S.; Li C.; Chen S.T.; Peng Z.Q.; Biological activity of the essential oil from the leaves of Piper sarmentosum Roxb. (Piperaceae) and its chemical constituents on Brontispa longissima (Gestro) (Coleoptera: Hispidae). Pestic Biochem Phys 2010, 96, 132-9, https://doi.org/10.1016/j.pestbp.2009.10.006.

  1. Ma Z.Q., Luan Z.C., Zhang X. Effects of Terpinen-4-ol on Culex pipiens Pallens and its Na+, K+-ATPase. Chin J Pestic Sci 2009, 11, 230-234.
  2. Leng X.F., Tang Z.H., Wang Y.C. Molecular Toxicology of Insecticides and Insect Resistance. China Agriculture Press 1993.
  3. Zhao S.H. Insect Toxicology. China Agriculture Press 1993.

Question 3. Acetylcholinesterase (AChE) is not a detoxification enzyme for xenobiotic substances, but a mode of action of some insecticides such as organophosphorus compounds.
Response: We appreciate very much for the reviewer’s valuable comments.

To make it clear, we rewrote it as: To explore whether the insecticidal function of the HK strain was related to the nervous system, i.e., the activity of AChE, which breaks down the neurotransmitter acetylcholine in the nervous system, was determined. After treatment with the crude extract, the AChE activity of adult A. craccivora was significantly higher 1.31-fold than the control group after 24 h (F1,4 = 7.242, P < 0.05; Fig. 3c). (P9. Line 338-342).

Question 4. In the biochemical tests, the crude extract of the LC50 concentration at 24h was used for the treatment. The survived aphids after 12, 24 and 48 h treatment were sampled. What about the mortality after 48 h treatment? If the mortality is very high, the biochemical results of 48 h treatment is not good for analyzing the biochemical effect of fungus extract on aphids. Because the survived insect under the high lethal pressure might be high tolerant, be dying with low physiological status, or with low fitness.
Response: We appreciate very much for the reviewer’s valuable comments. This is a very meaningful question and one that we need to explain. In our treatment, these survived aphids which have not Significant behavioral abnormalities were sampled by our observed.

Question 5. The title need to be changed. we cannot find any strong evidence for the "insecticidal mechanism".
Response: Thank you very much for your comment. We have revised the titlie “Insecticidal effect of entomopathogenic fungi Lecanicillium araneicola HK-1 in Aphis craccivora (Hemiptera: Aphididae)”

Question 6. Abstract & "5. Conclusions": must be rewritten. Some indications without any evidence cannot be in these two sections.
Response: We appreciate very much for the reviewer’s valuable comments. We have rewritten the Abstrace and Conclusion

Abstrace rewrote as: Aphis craccivora (Hemiptera: Aphididae) is an important pest affecting various crops worldwide. However, only a few studies exist on the infection and insecticidal mechanisms of A. craccivora by Lecanicillium. We investigated the infection process of A. craccivora with Lecanicillium araneicola HK-1 using fluorescence microscopy and scanning electron microscopy (SEM), and our results indicated that the conidia of strain HK-1 easily attached to the feet and dorsum of A. craccivora. Analysis of chitinase and extracellular protease activities indicated that they were induced by their substrates. A bioassay of A. craccivora showed that the median lethal mortality (LC50) of the crude extract was 24.00 mg mL-1 at 24 h. Additionally, the crude extract disrupted the detoxification enzyme system of A. craccivora, including inhibition of carboxylesterase (CarE) and induction of glutathione S-transferase (GST) and acetylcholinesterase (AChE). Gas chromatography-mass spectrometry (GC-MS) analysis suggested thatp-Cymene, Hymecromone, 9, 12-Octadecadienoic acid (Z, Z)- methyl ester, and 9, 12-Octadecadienoic acid (Z, Z) may be the insecticidal compounds, which lead to significant effects. This study provides a theoretical basis for the use of L. araneicola HK-1 as a potential biological control agent. (P1. Line 17-31)

Conclusions rewrote as: In summary, HK-1 strain has the capability to induced mortality in aphids A. craccivora. Treatment with the crude extract of HK-1 led to alterations in the enzymes (protease, chitinase, and ATPase) of the aphids as well as a considerable influence on the expression of genes associated with detoxification enzymes, predominantly inducing upregulation. Through gas chromatography-mass spectrometry (GC-MS) analysis, we identified substances that may be linked to the insecticidal effects on aphids. However, further experiments are required to isolate and confirm these specific active compounds. Our data provide a basis for further exploration of the potential use of strain HK-1 as a biocontrol agent against A. craccivora and to enhance the development of entomopathogenic fungi to control aphids and other harmful insects in agriculture. (P14.Line 492-501)

Question 7. 2.1: The origin and identification information, or the reference of the HK-1 strain should be provided.
Response: We appreciate very much for the reviewer’s valuable comments. We have provided this information (P3, Line 98-100: L. araneicola HK-1 was isolated from dead A. craccivora adults, and was deposited in the China Center for Type Culture Collection (CCTCC) as the strain code CCTCC M 2022252). And this source information and identification of the strain HK-1 have been published in our previous work Liu et al. 2022 (In Chinese), https://doi.org/10.16380/j.kcxb.2023.04.006.

Question 8. section 2.5-2.6: Replication settings should be described.
Response: Thank you very much for your comments. We have made it clear in the manuscript (P3-4, line 138-163).

2.5. The measurement of chitinase activity

For the measurement of chitinase activity, it was performed by DNS method[33]. 1 mL blastospores suspension (1×108 conidia mL-1) of strain HK-1 was inoculated in 100 mL different content colloidal chitin (0; 0.5%; 1%; 2%; 4% w/v) liquid medium (0.5% peptone; 0.05% KH2PO4; 0.05% KCl; 0.05% MgSO4·7H2O; 0.1% ZnSO4·7H2O), and then cultured at 28 ℃ on a shaker incubator (180 rpm) for 3, 4, 5, 6 and 7 d. The reaction mixture containing 0.5 mL of 1% (w/v) colloidal chitin and 1 mL enzyme solution was incubated at 50 ℃ for 1 h. The experiment was repeated three times. The released N-acetyl-D-glucosamine was measured spectrophotometrically at 540 nm. One unit of chitinase activity is defined as the mount of enzyme capable of digesting the colloidal chitin to produce 1µg of N-acetyl-D-glucosamine in 1 min.

2.6. The measurement of extracellular protease activity

For the measurement of extracellular protease activity, it was performed by Folin method[34]. 1 mL blastospores suspension (1×108 conidia mL-1) of strain HK-1 was inoculated in 100 mL different gelatin (0, 0.5%, 1%, 2%, 4% w/v) liquid medium (0.03% K2HPO4; 0.03% NaCl; 0.03% MgSO4·7H2O; 0.02 M phosphate buffer) at 28 ℃ on a shaker incubator (180 rpm) for 3, 4, 5, 6 and 7 d. The reaction mixture containing 1 mL of 2% (w/v) casein solution and 1 mL enzyme solution was incubated at 37 ℃ for 30 min. Then, the reaction was quenched for 20 min by adding 2 mL of trichloroacetic acid (0.4 mol L-1) at 37 ℃ water bath. The filtrate separated from the mixture by filtering through four layers of lens paper. Subsequently, the filtrate transferred into a 15 mL test tube and reacted with 5 mL of Na2CO3 solution (0.4 mol L-1) and 1 mL Folin Phenol reagent at room temperature for 5 min. Finally, the released tyrosine was measured spectrophotometrically at 680 nm. For each protease activity, the replication was three times. One unit of protease activity is defined as the mount of enzyme capable of digesting the casein to produce 1µg of tyrosine in 1 min.

Question 9. Provide the probit analysis software information.
Response: Thank you very much for your comment. This probit analysis software is SPSS 25.0 statistical software, and we added the information (Line 244-245)

Question 10. Table S3: there is only one data point below 50% mortality. This toxicity data is not good for the calculation of LC values.

Response: Thank you very much for your comment. We have added the date in the supplement file.

Table S3. The virulence test results of acetamiprid on A. craccivora adults at 48 h.

acetamiprid concentration (µg mL-1)

48 h average mortality ± SE (%)

Control

1.67 ± 0.03e

0.5

15.67± 0.03d

1

28.33 ± 0.03d

2

55 ± 0.10c

4

81.67 ± 0.03b

8

85 ± 0.05b

12

95 ± 0.00a

Different little letters in the table indicate significant differences between different treatments (P < 0.05).

Round 2

Reviewer 2 Report

Comments and Suggestions for Authors

The manuscript was much improved. However, I think some comments haven't been responded quite right.

For example, Q1: some misleading sentences can be still found in "Abstract".  L26-27: ‘Combined with Gas chromatography-mass spectrometry (GC-MS) analysis and bioassay tests,“ Which bioassay test? Bioassay results just showed two components had insecticidal activities. No bioassay test for these compounds in the current manuscript.

Q2: Although some references were listed in the author responding report,  there are many references to support "Na+, K+-ATPase plays a major role in the insect nervous system and excretory system as well as in the mammalian nerves, heart and kidney."

Q3: In "Abstract", "Materials and Methods", "Results" and "Discussion", the descriptions including AChE are still inappropriate. AChE is not a detoxification enzyme, even if this inappropriate and misleading description has been found in some references.

Q4: I keep my comment that the biochemical results of 48 h treatment are meaningless without the mortality data after 48 h treatment.

Q7: The reference of this source information and identification of the strain HK-1 should be added in section 2.1

Q8: Was the mortality data of 0.5  µg mL-1 acetamiprid treatment obtained in the same bioassay tests with 1, 2, 4, 8 and 12 µg mL-1 acetamiprid treatments? Or in a different test? If in the same test, why the mortality data of 0.5  µg mL-1 treatment was not included in the previous version of the manuscript. If not in the same test, these data should not been combined.

Author Response

Dear Reviewer2

Thank you for taking the time to review our manuscript Infection and insecticidal mechanism of Lecanicillium araneicola HK-1, Manuscript ID: insects-2639039. We have carefully considered your comments and made modifications accordingly. If you have any questions or concerns about the changes we have made, please let us know. We would be very grateful.

Thank you again for your time and expertise. I look forward to hearing from you.

Sincerely yours,

Dr. Pengfei Jin

School of Plant Protection, Hainan University, No. 58 Renmin Avenue, Haikou, Hainan, 570228, P.R. China

Email: jinpengfei@hainanu.edu.cn

The manuscript was much improved. However, I think some comments haven't been responded quite right.

For example, Q1: some misleading sentences can be still found in "Abstract".  L26-27: ‘Combined with Gas chromatography-mass spectrometry (GC-MS) analysis and bioassay tests,“ Which bioassay test? Bioassay results just showed two components had insecticidal activities. No bioassay test for these compounds in the current manuscript.

Response: Thank you very much for your comment.We have modified this inappropriate and misleading description in Abstract (P1. Line26-29: Combined with Gas chromatography-mass spectrometry (GC-MS) analysis, it is suggested that p-Cymene, Hymecromone, 9, 12-Octadecadienoic acid (Z, Z)- methyl ester, and 9, 12-Octadecadienoic acid (Z, Z) may be connected to the insecticidal effects.).

Q2: Although some references were listed in the author responding report, there are many references to support "Na+, K+-ATPase plays a major role in the insect nervous system and excretory system as well as in the mammalian nerves, heart and kidney."

Response: Thank you very much for your comment. In agreement with the reviewer’s valuable comments, we have modified this inappropriate description.

(P14. Line460-465): Ma studied the changes of Moina macrocopa when treated with Deltamethrin, and found after treated 24 h, Na+-K+-ATPase activity was reduced significantly[56]. And Leng et al. found that deltamethrin inhibited the Na+-K+-ATPase of housefly brain synaptosomes at concentrations ranging from 10-9 to 10-6. These study indicate that pesticide produce inhibitory effect of on Na+-K+-ATPase of the insects synaptosomes[57].

  1. 56. Ma Z.Q. Studies on the Relationship between Symptoms and Mechan of Different Kinds of Insecticides. Northwest Sci-Tech Uni-versity of Agriculture and Forestry. Yangling China. 2002.
  2. Leng X.F.; Xiao D.Q. Effect of deltamethrin on protein phosphorylation of houesfly brain synaptosome.∙Pestic∙Sci 1995, 44, 88-89. https://doi.org/10.1002/ps.2780440117.

(P14. Line471-475): In the present study, Ca2+-Mg2+-ATPase activity of A. craccivora was induced by the crude extract 48 h after treatment (Fig 5b). These changes may contribute to keep the relative stability of neuronal membrane potential, as we observed that even after 24 h of treatment with the crude extract A. craccivora was still can keep crawling. While further study need conduct to verify.

Q3: In "Abstract", "Materials and Methods", "Results" and "Discussion", the descriptions including AChE are still inappropriate. AChE is not a detoxification enzyme, even if this inappropriate and misleading description has been found in some references.

Response: Thank you very much for your comment. In agreement with the reviewer’s valuable comments, we have modified this inappropriate description in the whole manuscript.

In Abstract (P1. Line 24-26): Additionally, the results showed that the crude extract disrupted the enzyme system of A. craccivora, including inhibition of carboxylesterase (CarE), induction of glutathione S-transferase (GST) and acetylcholinesterase (AChE).

In Introduction (P2. Line 75-77): Insects metabolise harmful exogenous substances using enzymes system, including carboxylesterase (CarE) and induction of glutathione S-transferase (GST) and acetylcholinesterase (AChE) [25].

(P2. Line 84-88): Understanding the mechanism of the enzyme activity against the metabolites of strain HK-1 will be helpful for further applications.

In this study, we investigated the infection of A. craccivora with L. araneicola HK-1. We estimated mortality, enzyme activity, and enzyme-related gene expression in aphid adults after crude extract treatment.

In Materials and Methods (P4. Line181): 2.8. The effect of crude extract on enzyme activity.

(P5. Line 192): 2.9. The effect of crude extract on enzyme-related genes.

(P5. Line 214-216): To estimate Na+-K+-ATPase and Ca2+-Mg2+-ATPase activity, the treatment method is consistent with the description in ‘The effect of crude extract on enzyme activity’ earlier in this section.

In Results (P8. Line 322): 3.6. The effect of crude extract on enzyme activity.

(P10. Line 345): 3.7. The effect of crude extract on enzyme-related genes.

In Discussion (P13. Line 442-446): Insects produce enzymes, including CarE, GST, and AChE, to metabolise harmful exogenous substances[25]. The CarE structure included a α/β folded domain, an acyl binding domain, and a catalytic triad, resulting in an ability to hydrolyse ester bonds[26]. As an enzyme, an increase in the activity of GST is related to an increase in insect resistance[49].

(P14. Line 459-461): In summary, the results indicate that the crude extract disrupts the enzyme system of A. craccivora, leading to an impact on aphid physiological activities.

In Conclusions (P14. Line 490-493): In summary, HK-1 strain has the capability to induced mortality in aphids A. craccivora. Treatment with the crude extract of HK-1 led to alterations in the enzymes (protease, chitinase, and ATPase) of the aphids as well as a considerable influence on the expression of genes associated with enzymes, predominantly inducing upregulation.

Q4: I keep my comment that the biochemical results of 48 h treatment are meaningless without the mortality data after 48 h treatment.

Response: Thank you very much for your comment. We have checked and determined the mortality data (54.62%) after 48 h treatment. Actually, in the present study, we aim to explore the non-lethal effects of treating with L. araneicola HK-1 and its extracts, including alterations in enzyme activity and gene expression associated with enzymes. The findings revealed that even though the aphids did not die when treated with L. araneicola HK-1 and its extracts, there were changes in metabolism and some biochemical parameters. These findings are helpful in uncovering the potential pesticide mechanisms of L. araneicola HK-1.

Q7: The reference of this source information and identification of the strain HK-1 should be added in section 2.1

Response: Response: We appreciate very much for the reviewer’s valuable comments. We have provided this information (P3, Line 98-100: L. araneicola HK-1 was isolated from dead A. craccivora adults, and was deposited in the China Center for Type Culture Collection (CCTCC) as the strain code CCTCC M 2022252). And this source information and identification of the strain HK-1 have been published in our previous work[32].

Q8: Was the mortality data of 0.5 µg mL-1 acetamiprid treatment obtained in the same bioassay tests with 1, 2, 4, 8 and 12 µg mL-1 acetamiprid treatments? Or in a different test? If in the same test, why the mortality data of 0.5 µg mL-1 treatment was not included in the previous version of the manuscript. If not in the same test, these data should not been combined.

Response: We appreciate very much for the reviewer’s valuable comments.

Yes, the mortality data of 0.5 µg mL-1 acetamiprid treatment was obtained in the same bioassay tests with 1, 2, 4, 8 and 12 µg mL-1 acetamiprid treatments. We are very sorry for this mistake resulting from our inattention.

Round 3

Reviewer 2 Report

Comments and Suggestions for Authors

Thank authors for their careful revisions. I think this manuscript can be considered for publishing, only need to revise one description point.

P75-77 & P440-441: This is my third time to comment " Acetylcholinesterase (AChE) is not a detoxification enzyme for xenobiotic substances". However, in these sentences, "Insects produce enzymes, including ...AChE, to metabolise harmful exogenous substances"; "Insects metabolise harmful exogenous substances using enzymes system, including ... and acetylcholinesterase (AChE)". I suggest authors should read the classic textbook about "Insecticide toxicology".

Author Response

Dear Reviewer2

Thank you for taking the time to review our manuscript Infection and insecticidal mechanism of Lecanicillium araneicola HK-1, Manuscript ID: insects-2639039. We have carefully considered your comments and made modifications accordingly. If you have any questions or concerns about the changes we have made, please let us know. We would be very grateful.

Thank you again for your time and expertise. I look forward to hearing from you.

Sincerely yours,

Dr. Pengfei Jin

School of Plant Protection, Hainan University, No. 58 Renmin Avenue, Haikou, Hainan, 570228, P.R. China

Email: jinpengfei@hainanu.edu.cn

Reviewer2:

Thank the authors for their careful revisions. I think this manuscript can be considered for publishing, only need to revise one description point.

P75-77 & P440-441: This is my third time to comment " Acetylcholinesterase (AChE) is not a detoxification enzyme for xenobiotic substances". However, in these sentences, "Insects produce enzymes, including ...AChE, to metabolise harmful exogenous substances"; "Insects metabolise harmful exogenous substances using enzymes system, including ... and acetylcholinesterase (AChE)". I suggest authors should read the classic textbook about "Insecticide toxicology".

Response: Thank you very much for your comments. We accept your suggestion and have read the classic textbook and reference about insecticide toxicology which deepened our knowledge and awarenes about insecticide toxicology. We have modified this inappropriate and misleading description in the manuscript (P2. Line75-79: I The enzymes system, including carboxylesterase (CarE) and glutathione S-transferase (GST) inducted involve in insects metabolise harmful exogenous substances[25], and the enzymes acetylcholinesterase (AChE) can degrade acetylcholine, termi-nating the excitatory effect of neurotransmitters on the post-synaptic membrane, ensuring the normal transmission of neural signals in the body[25].) and (P13. Line441-443: Insects produce enzymes, including CarE and GST to metabolize harmful exogenous substances[25]. AChE is also induced to degrade acetylcholine, making an effort to ensure the normal transmission of neural signals[25].